# Mechanisms of Sustained Increases in *γ* Power Post-Ketamine in a Computational Model of the Hippocampal CA3: Implications for Ketamine’s Antidepressant Mechanism of Action

**DOI:** 10.3390/brainsci13111562

**Published:** 2023-11-07

**Authors:** Maximilian Petzi, Selena Singh, Thomas Trappenberg, Abraham Nunes

**Affiliations:** 1Faculty of Computer Science, Dalhousie University, Halifax, NS B3H 4R2, Canada; mx411426@dal.ca (M.P.); tt@cs.dal.ca (T.T.); 2Department of Psychology, Neuroscience & Behaviour, McMaster University, Hamilton, ON L8S 4L6, Canada; singhs11@mcmaster.ca; 3Department of Psychiatry, Dalhousie University, Halifax, NS B3H 4K3, Canada

**Keywords:** ketamine, computational model, neuron, depression, hippocampus, CA3

## Abstract

Subanaesthetic doses of ketamine increase γ oscillation power in neural activity measured using electroencephalography (EEG), and this effect lasts several hours after ketamine administration. The mechanisms underlying this effect are unknown. Using a computational model of the hippocampal cornu ammonis 3 (CA3) network, which is known to reproduce ketamine’s acute effects on γ power, we simulated the plasticity of glutamatergic synapses in pyramidal cells to test which of the following hypotheses would best explain this sustained γ power: *the direct inhibition hypothesis*, which proposes that increased γ power post-ketamine administration may be caused by the potentiation of recurrent collateral synapses, and *the disinhibition hypothesis*, which proposes that potentiation affects synapses from both recurrent and external inputs. Our results suggest that the strengthening of external connections to pyramidal cells is able to account for the sustained γ power increase observed post-ketamine by increasing the overall activity of and synchrony between pyramidal cells. The strengthening of recurrent pyramidal weights, however, would cause an additional phase shifted voltage increase that ultimately reduces γ power due to partial cancellation. Our results therefore favor the disinhibition hypothesis for explaining sustained γ oscillations after ketamine administration.

## 1. Introduction

Ketamine is an N-methyl-D-aspartate receptor (NMDAR) antagonist which, at subanaesthetic doses, induces rapid antidepressant effects in patients with treatment-resistant major depressive disorder (TRD) [1]. Therapeutic subanaesthetic doses of ketamine increase γ (30–100 Hz) oscillation power after acute administration [2,3,4,5,6,7]. Using a computational model of the cornu ammonis 3 (CA3) microcircuit, Neymotin et al. [8] showed that the NMDAR blockade of inhibitory oriens–lacunosum moleculare (OLM) cells (with or without direct blockade of the NMDARs at CA3 recurrent collateral synapses) is sufficient to observe ketamine-induced increases in γ power, but in their model, the change in γ power was resolved after the termination of the OLM NMDAR blockade. However, studies of patients with TRD receiving therapeutic subanaesthetic ketamine have found that these increases in γ power persist for several hours after application [9,10,11,12]. The relationship between γ oscillations and major depressive disorder (MDD) is complex, with studies comparing MDD patients to healthy controls employing heterogeneous definitions of the γ band, measuring across different brain regions and using task-based vs. resting-state paradigms (these studies are well reviewed by Fitzgerald and Watson [13]). Notwithstanding these variable baseline differences in MDD compared to healthy controls, there is consistent evidence that increases in γ power may be predictive of the clinical antidepressant response to subanaesthetic ketamine in TRD [9,10,12,14,15]. Cornwell et al. [10] and Nugent et al. [12] found a post-ketamine increase in γ power specifically in the cortical areas in patients who experienced at least a 50% reduction in symptoms with the treatment. de la Salle et al. [14] extended these results by finding that an increase in γ power in cortical areas 2 h after a single ketamine administration predicted a *sustained* antidepressant response following a standard therapeutic series of six infusions for three times a week over two weeks. In particular, large increases in γ power in both cortical and subcortical areas may be especially predictive of the treatment response in patients with lower preketamine levels of γ power [9]. Despite the complexity and diversity of γ power findings in MDD [13,16], the findings of these treatment studies together suggest that sustained increases in γ power following therapeutic subanaesthetic infusions may be an important biomarker of the treatment response, as well as a potential clue toward the physiology of depression and the mechanism of ketamine’s antidepressant action. The mechanisms by which ketamine induces this sustained increase in γ power are unknown. Therefore, identifying and understanding these mechanisms could shed light on the physiology of depression and ketamine’s antidepressant effect.

Ample evidence suggests that there is a negative link between depression and synaptic plasticity [17,18]. Depression has been associated with a reduction in the number of synapses and neurons and impaired synaptic functioning, as well as reduced levels of neurotrophic factors like brain-derived neurotrophic factor (BDNF) [18]. Successful treatments for major depressive disorder (MDD) such as selective serotonin reuptake inhibitors and electroconvulsive therapy have been shown to promote plasticity [19,20]. The antidepressant effect of ketamine in particular has been related to an increase in cortical excitability as measured by an increase in γ power in whole-head magnetoencephalography (MEG) data 230 min after infusion, which potentially connects the antidepressant effect with an increase in synaptic plasticity [10], as increases in γ power have been suggestive of an increase in synaptic plasticity [21]. Here, we consider synaptic strength increases based on the finding that ketamine increased α-amino-3-hydroxy-5-methyl-4-isoxazolepropionic acid receptor (AMPAR) miniature excitatory postsynaptic current (mEPSC) amplitudes in cultured hippocampal neurons by approximately 43% [22]. However, it is not clear by which mechanism the ketamine-related NMDAR blockade increases plasticity.

Previous hypotheses propose that ketamine-induced increases in plasticity may be attributed to the drug targeting different neuron types and thus engaging different plasticity mechanisms. There are two dominant hypotheses in this domain: the *disinhibition hypothesis* and the *direct inhibition hypothesis* [23], which the present study is primarily interested in testing. *The disinhibition hypothesis* posits that the NMDAR blockade on γ-aminobutyric acid (GABA)-ergic interneurons results in lower interneuron activity and, therefore, the higher activity of glutamatergic neurons, thus consequently inducing activity-dependent long-term potentiation (LTP) [23,24,25]. The *direct inhibition hypothesis* posits that the NMDAR blockade on glutamatergic neurons directly upregulates AMPAR expression locally in an activity-independent fashion at synapses where the NMDAR was blocked by ketamine [23,26]. The NMDAR blockade leads to the dephosphorylation of eukaryotic elongation factor 2 (eEF2), thereby leading to AMPAR protein translation upscaling [24,26,27].

The disinhibition and direct inhibition hypotheses imply different spatial distributions of plasticity along the somatodendritic tree. The CA3 model derived by Neymotin et al. [8] may help us study the effects of differential spatial distributions of plasticity on γ power changes post-ketamine. Specifically, Neymotin et al. [8] found that NMDAR antagonism on OLM cells could reproduce experimentally observed increases in γ power by disinhibiting pyramidal cells. Rate-based Hebbian plasticity rules [28] and the finding of heterosynaptic plastictiy in the CA3 [29] suggest that this disinhibition may then cause an increase of synaptic conductances via LTP for all active glutamatergic synapses in CA3 pyramidal cells (i.e., recurrent collaterals and synapses from areas outside of the CA3). Conversely, Neymotin et al. [8] also found that the direct inhibition of NMDARs in recurrent collateral synapses, in addition to those in OLMs, increased γ power during the active NMDAR blockade. Blockades of other (external) NMDARs at pyramidal neurons did not increase γ power. According to the direct inhibition hypothesis, such a spatially restricted NMDAR miniature current blockade would then cause eEF2 dephosphorylation restricted to the same area [26], thus implying that under the direct inhibition hypothesis of ketamine action, subsequent AMPAR upscaling in the CA3 would only occur at recurrent synapses.

It is unclear which of these plasticity mechanisms, locally at recurrent collaterals or more broadly throughout the somatodendritic tree, would best account for sustained γ power increases *after* ketamine-induced NMDAR blockades. The primary objective of the present study is to assess whether the disinhibition or direct inhibition hypothesis better explains sustained γ power increase post-ketamine administration in a previously established model of ketamine’s actions in the CA3 [8]. The model is comprised of multicompartmental Hodgkin–Huxley-style neurons. In this model, we will selectively alter plasticity at various locations of the pyramidal neuron somatodendritic tree, in line with the differing predictions of the disinhibition and direct inhibition hypotheses described above, to examine which of these manipulations can generate sustained increases in γ power, which would be expected post-ketamine administration.

## 2. Methods

### 2.1. Network Model

We employ an existing model of the hippocampal CA3 [8] (Figure 1), including three populations of neurons: pyramidal cells (*n* = 800, with somata, basilar one-compartment dendrites, and apical three-compartment dendrites), OLM (*n* = 200, with only somata), basket cells (*n* = 200, with only somata), and a medial septum, which paces the network with periodic inhibition. The rationale for parameterization governing cellular and network geometry, topology, biophysics, and background activity was reviewed by Neymotin et al. [8]. To maintain consistency with their model, we kept all parameters unchanged. Double exponential functions were used to model synaptic currents. Both the basket–basket connections and the basket–pyramidal connections were able to generate γ oscillations as has been previously described [30].

### 2.2. Cells

The neuronal geometry, topology, time constants, and conductances for synapses and background activity are chosen following Neymotin et al. [8]. Cellular dynamics were governed by sodium and delayed rectifier potassium currents, along with cell-type-specific mechanisms outlined below.

#### 2.2.1. Pyramidal Cells

The network includes 800 excitatory pyramidal neurons, with each including a soma; a short, one-compartment dendrite; and a longer, three-compartment dendrite. All compartments contain additional potassium type A currents for rapid inactivation, as well as a hyperpolarization-activated current for bursting [31].

#### 2.2.2. Oriens–Lacunosum Moleculare (OLM) Cells

The network includes 200 inhibitory OLM cells consisting of only a soma, which have high-threshold calcium and hyperpolarization-activated currents to facilitate bursting behavior, as in the original Neymotin et al. [8] model. Changes in calcium concentration were tracked to facilitate activation of calcium activated potassium current, which plays a role in sustained inactivation after bursting [32].

#### 2.2.3. Basket Cells

The 200 basket cells in the model each consist of a singular compartment representing the soma. Simulated physiology is partially governed by a transient sodium and delayed rectifier potassium current.

### 2.3. Connectivity

#### 2.3.1. Connections

Network connectivity is illustrated in Figure 1. Only the basket cell and pyramidal cell populations have within-population connections.

#### 2.3.2. Synapses

Pyramidal neurons activate AMPARs and NMDARs of their postsynaptic target cells, while Basket and OLM cells activate ${\mathrm{GABA}}_{A}$ receptors of their targets. All synapses are modeled by a double exponential current. As in Neymotin et al. [8], an additional synaptic delay of 2 ms was introduced to model the propagation of the signal along the axon and synapse. Synaptic time constants are listed in Table 1.

#### 2.3.3. Background Activity

Poisson-distributed spikes were sent into the network to synapses at somata and dendrites (at connections marked by “ext” in Figure 1, but also connections targeting OLM and basket cells) according to parameters outlined in Table 2. A 1000 Hz Poisson-distributed input was sent to AMPA and GABA receptors on the somata of all cells to model the high conductivity state caused by the high number of inputs to each neuron found in vivo [33]. In addition, a 10 Hz Poisson-distributed input targeting the NMDA receptors in the distal compartment of the apical dendrite of pyramidal cells [34] models the input from the entorhinal cortex.

A medial septum (MS) theta drive is included here as a “pacemaker” current, as the MS is classically thought to drive the hippocampus at theta frequency [35]. It periodically (every 150 ms) inhibits all basket and OLM cells by applying a double exponential current with a rise time of 20 ms, a fall time of 40 ms, and a reversal potential of −80 mV [8].

### 2.4. Experimental Conditions

#### Comparison of Effects of Direct Inhibition and Indirect Disinhibition on γ Power

Ketamine application may upscale AMPAR conductances in pyramidal cells, either by increasing the recurrent collateral and external synaptic weights (following the disinhibition hypothesis) and/or by upscaling the recurrent collateral glutamateric synapse (following the direct inhibition hypothesis).

We therefore simulated 7000 ms recordings from the CA3 model under various degrees of AMPAR conductance scaling at recurrent collateral (multiplicative scaling factor $k_{rec}\geq 1$) and external pyramidal synapses at somata and apical dendrites (multiplicative scaling factor $k_{ext}\geq 1$). In numerical simulations of biophysical networks, there is often a period of time initially where the network’s behavior is not yet stable and is also still sensitive to the initial conditions. To avoid this, we discarded the first 3000 ms of each simulation to allow for equilibration. The control condition (i.e., preketamine) was represented as $k_{ext}=k_{rec}=1$. Connections and background activity for control and ketamine simulations were initialized with the same random seed per run. The AMPAR upscaling attributable to ketamine was found to be approximately 43% [22]. To prevent an under-estimation of the scaling factor, we chose the set of tested scaling factors to be $\left\{\left(k_{rec},k_{ext}\right)|k_{rec},k_{ext}=1,1.25,1.5,1.75\right\}$. The blockade of NMDA receptors, as modeled in Neymotin et al. [8], was not modeled in the present study, since the focus of our study was the post-ketamine effects.

Local field potentials (LFPs) were computed as in Neymotin et al. [8] by taking the difference between the voltage at the most distal and the basal dendrite of each pyramidal neuron and averaging this difference over all pyramidal neurons. Spectral power of the resulting time series was calculated with the Welch’s method implemented in Python (version 2.7.18) with the module Scipy, and cumulative LFP-γ power was computed by summing these spectral power values between 30–100 Hz. Henceforth, it is this LFP γ power which we denote simply as γ power. For the comparison of direct inhibition and indirect disinhibition, we then computed the relative change in γ power, $\Delta_{\gamma }$, for condition $\left(k_{rec},k_{ext}\right)$ relative to the control $k_{ext}=k_{rec}=1$.

We evaluated the effects of $k_{ext}$ and $k_{rec}$ on $\Delta_{\gamma }$ using the following linear regression model, $\Delta_{\gamma }∼k_{ext}+k_{rec}$ (in R notation), using the R statistical programming language (R version 4.2.3).

Many spiking patterns can generate the same LFP [36], which makes conclusions about the network’s spiking behavior difficult to form. To better understand the fundamental mechanism behind our results, we also considered γ power of the cumulative spiking activity of the network (which we will call “raster γ power”), in addition to γ power of the LFP. This raster γ power was calculated by combining the spike times of all neurons into to one single vector, then performing convolution of a 5 ms square kernel over this vector. Finally, raster–γ power was computed similarly to LFP–γ power.

In order to identify factors that explain why synaptic weight scaling (via $k_{rec}$ and $k_{ext}$) alter γ power, we conducted mediation analyses with the R package mediation (Figure 2) after standardizing each variable in the data. Mediation analysis assumes an independent variable *X* (here, either the scaling factor $k_{rec}$ or $k_{ext}$), a dependent variable *Y* (here, γ power) and a mediator *M*. We hypothesized that γ power increases were mediated by pyramidal cell firing rates, and so average pyramidal cell firing rate was set as the mediator variable.

## 3. Results

### 3.1. Comparison of γ Power Effects in Models of Direct Inhibition and Indirect Disinhibition

The network behavior is shown in raster plots in Figure 3 for (A) the baseline condition, (B) the condition of increased external weight strength, and (C) the condition of (strongly) increased recurrent weight strength. The network fell into seizures for $k_{rec}$ values higher than 37; therefore, we chose 37 as the value for this parameter.

Results demonstrating the effects of recurrent and external synaptic scaling are shown in Table 3. γ power was increased only through the scaling of the AMPA receptor conductances on the pyramidal cell synapses receiving external inputs ($\beta_{k_{ext}}=1.17,p<0.001$, Table 3). Scaling the AMPA receptor conductances at recurrent collateral glutamatergic synapses did not induce sustained γ power increase in our model ($\beta_{k_{rec}}=0.13,p=0.362$). Table 4 indicates that the effect of external synaptic scaling via $k_{ext}$ on γ power was fully mediated by increases in the pyramidal cell firing rates.

### 3.2. Explanation for Why $k_{rec}$ Upscaling Does Not Impact γ

Conductances at recurrent synapses are low in comparison to external synapses (compare the respective synapses in Table 2 and Table 1), which might explain why scaling them up or down has only little effect on the network’s behavior. However, a simple comparison of the conductances is hardly convincing, as the effect of recurrent conductance scaling on the network behavior might be much stronger than that of external weight scaling, due to potential positive feedback loops through recurrent connections. Furthermore, we sought to ensure that our results were independent of our chosen scaling range of 1 to 1.75 and of the chosen default recurrent weight strength. To further investigate the mechanism behind the missing γ increase with the $k_{rec}$ increase, we additionally ran simulations for higher than plausible scaling factors. The results can be seen in Figure 4b.

Figure 4 shows that the pyramidal firing frequency was increased with stronger recurrent connections, which was similar to the condition with stronger external connections. However, LFP–γ power decreases. We hypothesized that the recurrent connections at the pyramidal neurons would be unable to increase γ power measured at the pyramidal neurons because of the following effect: After a pyramidal population spike, the voltage increase caused by the recurrent connections arrives phase shifted by half of a γ cycle compared to the extant γ oscillation. This would correspond to an increase in the potential measured at the pyramidal somata at γ trough rather than the peak, thereby causing an overall reduction in γ amplitude. This reduction in the observed γ amplitude corresponds to a reduction in the measured γ power.

Our proposed explanation for the negative effect of the recurrent connection strength on LFP–γ power predicted that delaying the time of the voltage increase after a spike by half of a γ cycle would invert the negative effect of the $k_{rec}$ on γ power, and the synaptic delay of a full γ cycle would recover the negative effect of recurrent connectivity on γ power. The following section tests this hypothesis.

#### γ−krec Relationship Depends on Timing of Population Spike Arriving through Recurrent Connections

A γ cycle here is approximately 33 ms long (the peak in the Fourier spectrum is approximately 33 Hz). Figure 5 confirms that increasing the synaptic delay by approximately half a γ cycle (17 ms) inverted the effect of the $k_{rec}$ on γ power, a delay of a whole γ cycle recovered the negative effect, and a delay of one and a half γ cycles again produced a positive effect. Note that increasing the recurrent conductances beyond a certain point caused an abrupt phase transition of the network into a state comparable to epileptic seizures, with very high firing rates (see Figure 4b). Here, we only analyzed the nonepileptic regime.

To illustrate the voltage increase timing under normal conditions, we measured the voltage before and after a population spike with no added synaptic delay. Figure 6 shows the voltage at a pyramidal soma after spikes were sent to the basal dendrite (similar to recurrent pyramidal connections) for an otherwise silent network. The voltage curve indicates that, under normal conditions, the majority of the postsynaptic voltage increase caused by recurrent connection took place between the γ peaks. When a large enough synaptic delay was added, shifting this curve to the right, much of the effect of one γ peak could affect the next γ peak, which increased LFP–γ power, as opposed to at the trough.

As shown in Figure 4b, raster–γ power increased with the $k_{rec}$ value, while LFP–γ power decreased. While this may seem paradoxical, a potential explanation for why raster–γ power increased while LFP–γ power decreased is that the voltage increase during γ trough caused by recurrent connections does not increase the number of spikes, but rather only increases the subthreshold membrane potential to a degree that nonetheless impacts γ power as measured from the LFP. This impact on γ power within the LFP is because LFP–γ power takes subthreshold membrane potential oscillations into account and not just the spiking behavior, which is exclusively captured by raster–γ power. In fact, it can be seen in Figure 3 that the network with high $k_{rec}$ values produced almost no spikes during γ troughs. Therefore, the increase (as opposed to the decrease or stability as seen in LFP–γ power) of raster–γ power with increasing $k_{rec}$ values can be explained simply by the increase in the pyramidal firing frequencies.

## 4. Discussion

The mechanism behind the sustained γ power increase after the NMDA receptor antagonism with subanaesthetic ketamine infusion currently remains unclear, with two competing putative mechanisms considered to be at play: the disinhibition hypothesis and the direct inhibition hypothesis [23].

Previous modeling studies have suggested that the disinhibition hypothesis, whereby ketamine preferentially blocks NMDARs at inhibitory interneurons, explains the γ power increase during the *acute* application of ketamine. The resulting increased pyramidal activity should induce activity-dependent plasticity on recurrent collateral and external glutamatergic synapses. The present study extends these results by demonstrating that this activity-dependent plasticity at external and glutamatergic synapses is likely responsible for sustained increases in γ power post-ketamine. Under the assumption that the results from our CA3 model would also apply to cortical microcircuits, our data would support the disinhibition hypothesis as the source of γ power increases that have been repeatedly demonstrated following therapeutic ketamine administration for patients with TRD [9,10,11,12]. Notwithstanding, we do not claim that the disinhibition hypothesis describes the sole mechanistic pathway for ketamine’s antidepressant effects. There is evidence that direct inhibition of the NMDARs on glutamatergic neurons should induce local activity-independent plasticity, which in our model would occur only on recurrent collateral synapses. This local activity-independent plasticity is mediated by eEF2 dephosphorylation in close proximity to antagonized NMDARs [26]. This eEF2 kinase inhibition is able to produce antidepressant effects [37]. Finally, ketamine is hypothesized to have several other mechanisms of action aside from its activity at the NMDARs, including the modulation of opioid, monoamine, and cholinergic systems, as well as activity at the σ receptors and GABA${}_{A}$ receptors [38]. Therefore, while our results offer some support for the disinhibition hypothesis as an explanation of plasticity-mediated increases in γ power post-ketamine, future studies should seek to model this phenomenon in other microcircuit and broader brain system models with the incorporation of ketamine’s other known mechanisms of action.

We found that sustained γ power increase in a CA3 microcircuit required increasing conductances at the pyramidal cell glutamatergic synapses that receive input from outside of the CA3 (i.e., *external* synapses). This suggests that ketamine-induced disinhibition, but not direct inhibition, is sufficient to explain the sustained γ power increase, at least in the CA3 (further studies in cortical microcircuit models would be required to predict whether these results may hold in cortical areas). We found two reasons for the different effects on γ power of external and recurrent connections to the CA3 in our model. First, recurrent connections increase subthreshold membrane potentials at pyramidal somata during γ trough, which decreases γ power in opposition to γ power that would have been induced purely by spike synchrony at γ frequencies. Second, multiplicatively scaling recurrent conductances has generally small effects (in the nonepileptic regime) due to the small baseline conductance of these weights. In light of these prior data, our study offers greater support for the disinhibition hypothesis as the underlying mechanism of sustained γ power increase after ketamine-induced NMDAR antagonism. Therefore, further study of the detailed mechanisms underlying disinhibition-induced plasticity after ketamine administration may shed light onto its antidepressant actions. Gaining a better understanding of ketamine’s mechanism of action might help inform the development of pharmacological treatments for depression that are highly specific and minimize adverse effects. Our findings also emphasize the importance of studying other novel treatments that may act by increasing excitability and plasticity. These treatments include classical psychedelics such as psilocybin, which increase excitability by activating serotonin 5-HT2A receptors (5-hydroxytryptamine receptor 2A) [24], as well as high-frequency repetitive transcranial magnetic stimulation to the left dorsolateral prefrontal cortex [39].

A number of experimental tests can be conducted to confirm our predictions, in particular if we have controlling access to the nonrecurrent, external synapses targeting pyramidal cells. We make the following predictions:The stimulation of excitatory pathways targeting the CA3 increases pyramidal firing frequency and γ power. This is testable with hippocampal slice recordings;The lesions of external inputs to the CA3 should decrease γ power and firing frequency directly, as well as diminish γ power increases post-ketamine;An increase in pyramidal cell activity caused by ketamine is enough to cause LTP of pyramidal cells targeting synapses. This would be testable with optogenetic stimulation of parts of the CA3 and by comparing the results of population excitatory postsynaptic potential slope recordings before and after stimulation, similar to previous LTP experiments [40];Ketamine increases the connection strength from external areas to the CA3. This would be indirectly testable through LFP recordings in the CA3 and afferent areas, as well as directly via slice stimulation/recording paradigms;Any external stimulation of the CA3 increases firing frequency and γ power in the long term. This should be testable with optogenetic stimulation, as well as recordings of LFPs and single units.

In addition, further experiments should use causal mediation analyses to confirm that a γ power increase predicts the antidepressant response to ketamine. To obtain the data for this mediation analysis, electroencephalography (EEG) in combination with measures of antidepressant effect could be implemented alongside randomized controlled trials of ketamine vs. a suitable control agent. One may also consider evaluating CA3 functioning indirectly using behavioral tasks in patients receiving ketamine [41]. Cued recall tests may serve as a putative probe of pattern completion abilities [41], which is a neural computation that occurs when a partial cue reactivates a complete representation [42,43]. This computation has classically been linked to the CA3 by computational modelers [44]. Building a computational model of the CA3 that is capable of performing this computation while simultaneously capturing the impacts of ketamine on γ power, with parallel cued recall behavioral and EEG data from humans, may help us gain a more cohesive understanding of ketamine’s neurocomputational and behavioral impacts, as well as how those are related to changes in power. More specifically, such a series of studies may help us understand what specific neurocomputational impacts that increases in γ power are indicative of.

Our model is based on Neymotin et al. [8], where the parameters have been tuned to reproduce healthy network behavior. In depression, the hippocampal volume is reduced [45], the left CA3 volume is decreased [46], and the activity of the left hippocampus and parahippocampal gyrus is increased [47]. Our network, therefore, does not precisely model ketamine application in the CA3 as it would definitively exist in patients with MDD. That being said, these differences in the CA3 network structure and function between MDD and healthy controls do not provide precise guidance on the parameter selections for a microcircuit model as we have implemented here. Acknowledging this limitation, we simply used an already existing model for the healthy CA3. Future research on the cellular and micro-circuit properties of the CA3 in MDD would help develop more precise and generalizable computational models of neural microcircuit functioning in MDD. One possibility here includes studies of induced pluripotent stem cells (iPSCs) derived from patients with MDD, which is similar to what has been implemented in studies of bipolar disorder [48,49,50,51]. Studying their electrophysiology and gene expression both before and after in vitro ketamine application may help develop a more comprehensive picture of the cellular mechanisms and the parameters for a CA3 model of patients with MDD.

Another limitation of our study is the primary focus on the CA3 microcircuit. While studies finding increased γ power post-ketamine have only reached cortical depth in humans, it is not clear that this translates to the hippocampus. In rodents, however, a parallel increase in power post-ketamine in both the prefrontal cortex and hippocampus has been observed [52]. Indeed, this model was adopted based on its demonstrated ability to explain ketamine-induced γ oscillations in animals [8]. This allowed us to simply extend the model to examine sustained oscillations. Unless post-ketamine γ power increases are found in the hippocampus in humans, future studies should investigate the electrophysiological signatures of the disinhibition and direct inhibition hypotheses in cortical models with validation against clinically obtained EEG data.

Following the original implementation of the CA3 model by Neymotin et al. [8], our model includes both NMDA and AMPA receptors, and we upscaled the AMPA receptors to simulate the effects of ketamine. However, it should be noted that the impact of the AMPA receptors on γ oscillations is not necessarily straightforward [53], and our model is naturally very simplified. For example, the study conducted by Li et al. [53] demonstrated that the application of the synthetic glutamate analog AMPA to CA3 slices ex vivo led to the downregulation of γ oscillation power, but it increased the peak oscillation frequency, and they linked this effect to the modulation of an intracellular signaling pathway impacting neuronal excitability [53]. Although this study highlights that AMPAR activation will not necessarily lead to γ power upregulation *in all cases*, it should be noted that AMPA is not found endogenously to begin with. Studies conducted in humans and rodents have, however, demonstrated that the AMPAR blockade and genetic manipulation of the AMPARs both lead to the downregulation of γ oscillations, respectively [54,55], thus highlighting the importance of AMPARs in γ oscillation regulation in vivo. Nonetheless, γ up- or down-regulation is linked to differential intracellular signaling pathway activation that is ligand-dependent, and, thus, studying those mechanisms would require a much more detailed model to capture how the biochemistry interacts with neuronal biophysical behavior. Although beyond the scope of our study, we believe this would be an interesting and valuable approach for teasing apart these nuances in future studies.

It is also worth noting that AMPARs are not the only receptors that regulate γ oscillations: potassium and calcium channels have also been shown to play a large role by modulating neuronal intrinsic excitability [56]. Understanding how ketamine may potentially interact with those channels will help us gain a more comprehensive understanding of how γ power is up-regulated after ketamine administration, which is another area for future work.

Finally, our study is limited by an exclusive focus on ketamine’s NMDAR-related mechanism of action. To more comprehensively understand ketamine’s antidepressant effects, other mechanisms such as long-term structural plasticity might play a role [24], in addition to effects on other neural systems [38]. Differentiating between ketamine metabolites may also be important in understanding the antidepressant effect, thus yielding evidence of an effect that is independent of NMDAR antagonism [57]. Another potential mechanism responsible for ketamine’s antidepressant action might be its anti-inflammatory action [58]. While our findings favor the disinhibition hypothesis of ketamine to explain the sustained γ power increase, it should be kept in mind that, due to our assumptions, other models might find results favoring other mechanisms.

## 5. Conclusions

In conclusion, we assessed whether the plasticity related to the disinhibition or direct inhibition hypothesis better explains the sustained γ power increases post-ketamine administration [23]. We used a previously established computational model of the hippocampal CA3 [8] that is capable of simulating the effects of ketamine on oscillatory dynamics to investigate these two hypotheses. Our results favor the disinhibition hypothesis, thus suggesting that increases in γ power post-ketamine may be attributable to the potentiation of external synapses onto pyramidal neurons. Our results highlight the importance of neuronal synchrony, and they emphasize the role of AMPAR-mediated currents and plasticity in promoting γ oscillations. Predictions from our modeling work can help inform future research into ketamine’s mechanism of action to help push research on MDD’s pathogenesis and treatment forward. 

## Figures and Tables

**Figure 1 brainsci-13-01562-f001:**
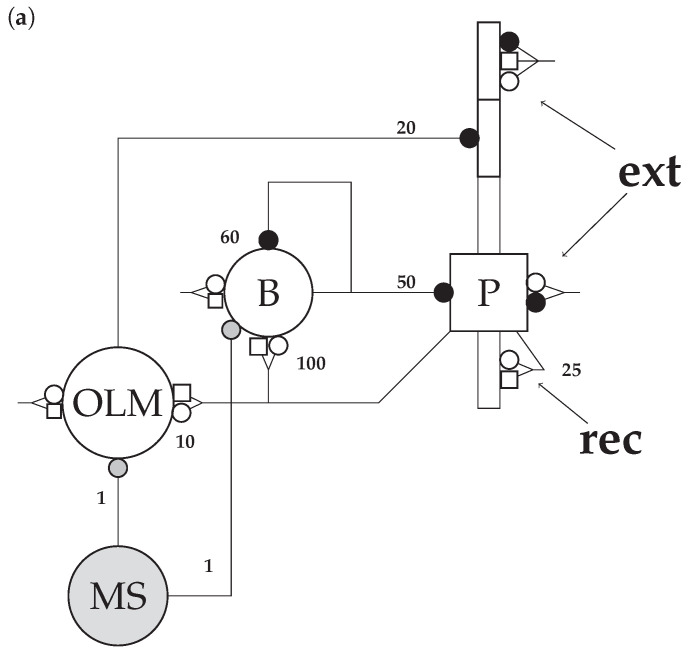
The cornu ammonis 3 (CA3) model. Figure adapted from Neymotin et al. [8]. *Abbreviations:* B (basket cells), P (pyramidal cells), OLM (oriens–lacunosum moleculare cells), medial septum (MS). (**a**) Open circles denote α-amino-3-hydroxy-5-methyl-4-isoxazolepropionic acid receptors (AMPARs), open squares denote N-methyl-D-aspartate receptors (NMDARs), and closed circles denote (inhibitory) γ-aminobutyric acid (GABA) receptors. The numbers on the connections are convergence numbers representing the number of randomly picked presynaptic neurons that are connected to each postsynaptic neuron. The locations of action of the scaling factors are marked as “ext” (for external synapses, which are scaled by $k_{ext}$) and “rec” (for recurrent collateral synapses, scaled by $k_{rec}$). (**b**) Higher level depiction of the same network under the different conditions that will be modeled in the present. Red diamonds mark locations where NMDARs are blocked during ketamine administration. Green circles mark locations where AMPAR upscaling takes place post-ketamine as a consequence of prior NMDAR blockade.

**Figure 2 brainsci-13-01562-f002:**
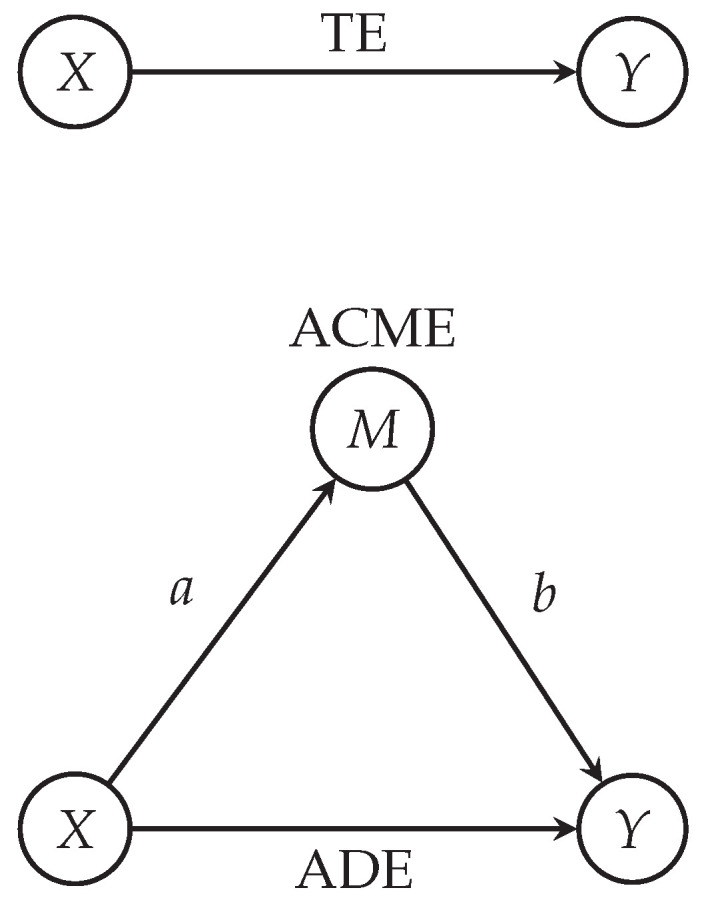
Illustration of mediation model. Mediation analysis evaluates the mechanism by which an independent variable (here, *X*) affects a dependent variable (here, *Y*) through an intervening or mediator variable (here, *M*). *a* is the path from *X* to *M*, thus representing the relationship between the independent variable and the mediator. *b* is the path from *M* to *Y*, thus representing the relationship between the mediator and the dependent variable after accounting for the influence of the independent variable. ADE (average direct effect) is the path from *X* to *Y* when the mediator is taken into account, thus representing the average effect of *X* on *Y* independent of the mediator. TE (total effect) is the effect of *X* on *Y* without considering the mediation by *M*. The TE combines both direct and indirect effects (through mediator). ACME (average causally mediated effect) is the effect of *X* on *Y* via the mediator *M*, which is represented by the path ab from X→M→Y.

**Figure 3 brainsci-13-01562-f003:**
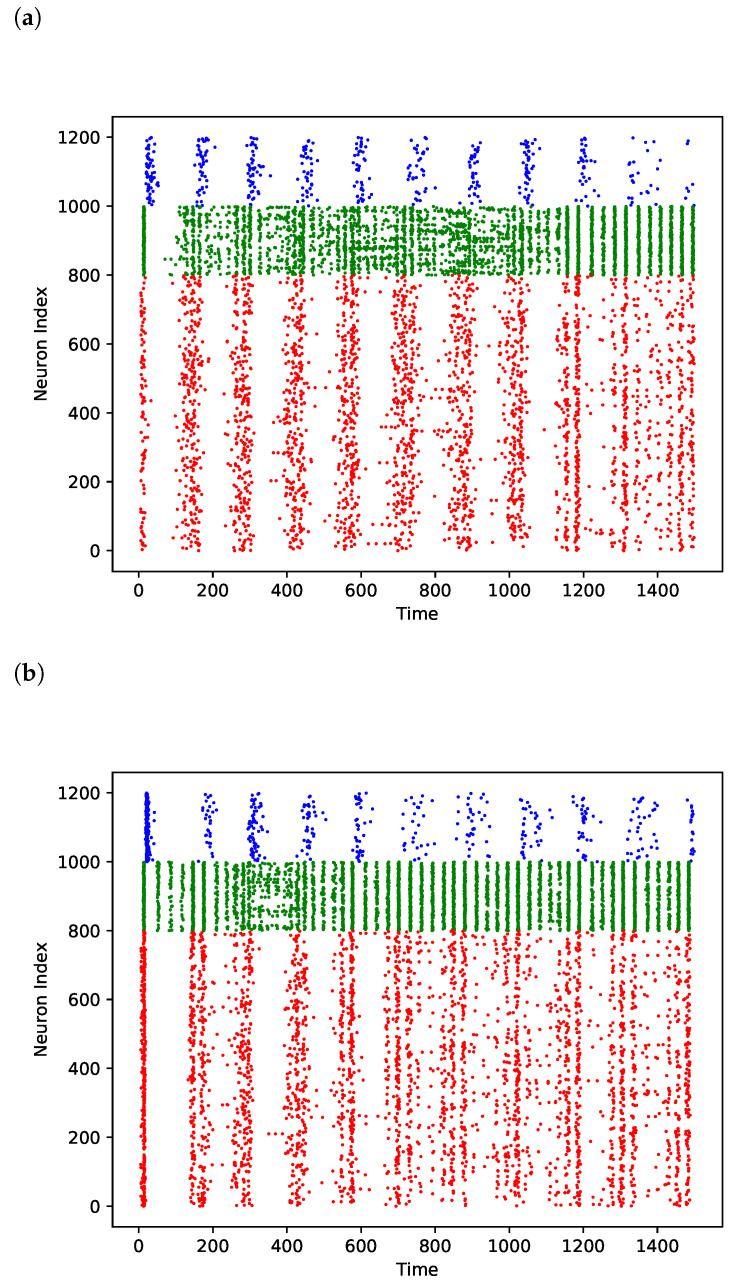
Spike times of OLM neurons (blue), basket cells (green), and pyramidal neurons (red) of network under different conditions. Horizontal axis is time in milliseconds, neurons are vertically arranged by index. (**a**) Normal conditions ($k_{rec}=1,k_{ext}=1$). (**b**) Increased external conductivity ($k_{rec}=1,k_{ext}=1.75$). (**c**) Increased recurrent connection strengths ($k_{rec}=37,k_{ext}=1$).

**Figure 4 brainsci-13-01562-f004:**
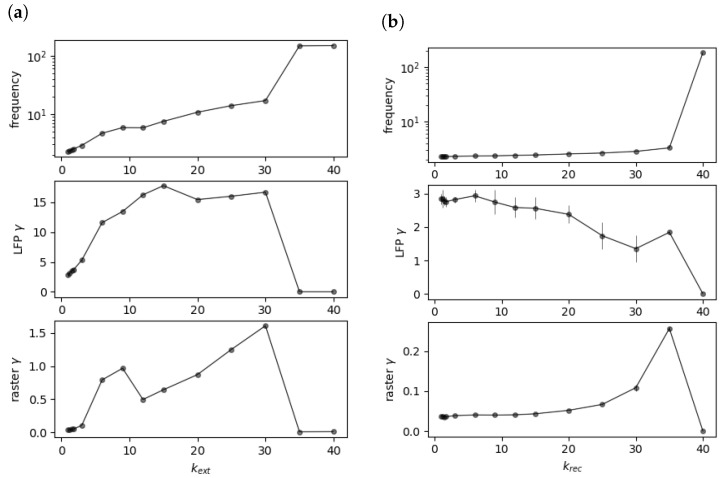
Resulting pyramidal spiking frequencies (average over all neurons), LFP–γ power, and raster–γ power for recurrent connection strengths of $k_{ext}$ (**a**) and $k_{rec}$ (**b**). Error bars are variances.

**Figure 5 brainsci-13-01562-f005:**
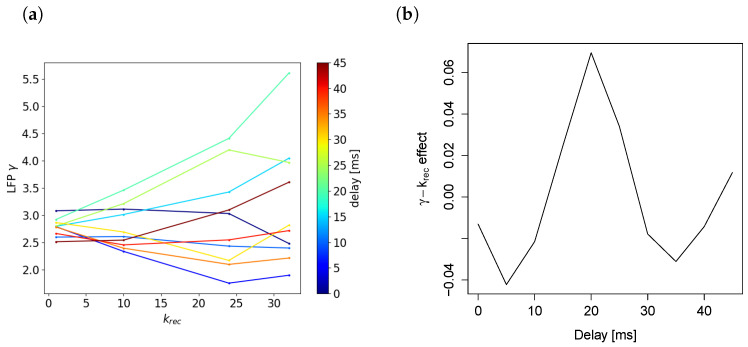
γ power−$k_{rec}$ relationships for different synaptic delays added to recurrent connections. The slope depends on the delay with respect to the length of a γ cycle (34 ms). (**a**) Average over 8 simulations of γ power plotted against $k_{rec}$ for different delays. (**b**) Fitted slope of the lines in (**a**) plotted against delay time.

**Figure 6 brainsci-13-01562-f006:**
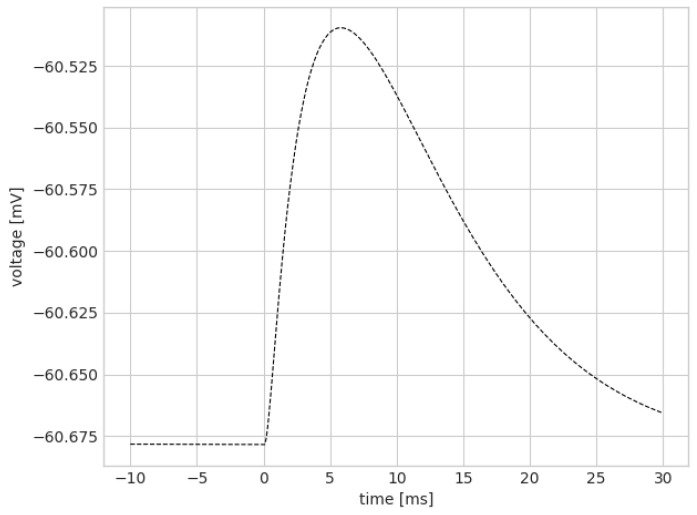
Voltage at one pyramidal soma before and after sending a spike to AMPA receptors of recurrent connections of all pyramidal neurons at t=0 ms, with recurrent conductances reduced to 10%. All other inputs to pyramidal neurons were silenced. The voltage curve is the result of double exponential synapses and dendritic filtering, but does not include the default synaptic delay of 2 ms, which would shift the curve to the right by 2 ms.

**Table 1 brainsci-13-01562-t001:** Synaptic parameters between populations. $\tau_{1}$ [ms]: rising time constant; $\tau_{2}$ [ms]: falling time constant; g: conductance. Note that all connections here target somata, except for the OLM–P connection, which targets the second apical dendrite. Values are set as in Neymotin et al. [8].

Presynaptic	Postsynaptic	Receptor	$\tau_{1}$ [ms]	$\tau_{2}$ [ms]	*g* [nS]
Pyramidal	Pyramidal	AMPA	0.05	5.3	0.02
Pyramidal	Pyramidal	NMDA	15	150	0.004
Pyramidal	Basket	AMPA	0.05	5.3	0.36
Pyramidal	Basket	NMDA	15	150	1.38
Pyramidal	OLM	AMPA	0.05	5.3	0.36
Pyramidal	OLM	NMDA	15	150	0.7
Basket	Pyramidal	${\mathrm{GABA}}_{A}$	0.07	9.1	0.72
Basket	Basket	${\mathrm{GABA}}_{A}$	0.07	9.1	4.5
OLM	Pyramidal	${\mathrm{GABA}}_{A}$	0.2	20	72
MS	Basket	${\mathrm{GABA}}_{A}$	20	40	1.6
MS	OLM	${\mathrm{GABA}}_{A}$	20	40	1.6

**Table 2 brainsci-13-01562-t002:** Parameters for background activity. $\tau_{1}$ [ms]: rising time constant; $\tau_{2}$ [ms]: falling time constant; g: conductance; Adend3: third compartment of the apical pyramidal dendrite. Values set as in Neymotin et al. [8].

Cell	Section	Synapse	$\tau_{1}$ [ms]	$\tau_{2}$ [ms]	*g* [nS]
Pyramidal	Soma	AMPA	0.05	5.3	0.05
Pyramidal	Soma	${\mathrm{GABA}}_{A}$	0.07	9.1	0.012
Pyramidal	Adend3	AMPA	0.05	5.3	0.05
Pyramidal	Adend3	NMDA	15	150	6.5
Pyramidal	Adend3	${\mathrm{GABA}}_{A}$	0.07	9.1	0.012
Basket	Soma	AMPA	0.05	5.3	0.02
Basket	Soma	${\mathrm{GABA}}_{A}$	0.07	9.1	0.2
OLM	Soma	AMPA	0.05	5.3	0.0625
OLM	Soma	${\mathrm{GABA}}_{A}$	0.07	9.1	0.2

**Table 3 brainsci-13-01562-t003:** Regression table for the effect of the scaling factors on γ power. *Abbreviations:* 95% confidence interval (CI), scaling factor for external ($k_{ext}$), and recurrent collateral synapses ($k_{rec}$).

Predictors	Estimates (β)	CI	*p*
(Intercept)	−1.30	−1.86–−0.74	<0.001
$k_{rec}$	0.13	−0.15–0.42	0.362
$k_{ext}$	1.17	0.89–1.46	<0.001
Observations	256		
$R^{2}$/$R^{2}$ adjusted	0.209/0.203		

**Table 4 brainsci-13-01562-t004:** Results of mediation analysis of the effect of $k_{ext}$ on γ mediated by pyramidal firing frequency. Values for $k_{ext}$ were 1, 1.25, 1.5, and 1.75, with each having 116 samples. CI: Lower and upper 95% confidence interval boundaries. ACME: average causal mediation effects. ADE: average direct effects.

Variable	Estimate	CI	*p*-Value
ACME	0.608	0.416–0.81	<$2\cdot 10^{-16}$
ADE	0.105	−0.104–0.30	0.32
Total Effect	0.713	0.645–0.77	<$2\cdot 10^{-16}$
Prop. Mediated	0.850	0.582–1.15	<$2\cdot 10^{-16}$

## Data Availability

The code to generate the data and figures is available from the following: https://github.com/MaximilianPetzi/ketamine_gamma_CA3 (accessed on 1 November 2023).

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
