# Peer review of "Mechanisms of Sustained Increases in γ Power Post-Ketamine in a Computational Model of the Hippocampal CA3: Implications for Ketamine’s Antidepressant Mechanism of Action"

_brainsci, 2023, doi:10.3390/brainsci13111562_

Round 1

Reviewer 1 Report

Comments and Suggestions for Authors

1. While the introduction provides background information, it lacks a clear, concise thesis statement that summarizes the main purpose and focus of the research. The disinhibition and direct inhibition hypotheses are introduced somewhat abruptly.

2. The introduction mentions previous studies, however, it could benefit from providing more context and background information on the significance and implications of gamma (γ) power increases and its relationship with depression and ketamine treatment.

3. The results mention unexpected or paradoxical findings but could benefit from more detailed explanations and discussions of these points

4. The discussion does not clearly elucidate the specific processes through which ketamine interacts with NMDA receptors.

5. The discussion seems to lean more towards supporting the disinhibition hypothesis without providing a balanced view or adequate comparison with the direct inhibition hypothesis.

6. The behavioral correlates of CA3 functioning in patients receiving ketamine are mentioned but not elaborated upon. Please explain these correlates in depth.

7. Emphasize the potential broader impacts of the research, including its relevance to depression treatment and how the study could pave the way for more targeted therapeutic approaches. 

8. The description of network connectivity and synaptic parameters is thorough and well-explained. However, providing a brief rationale for the chosen values or references to where they were derived from would add context. 

9. The explanation of experimental conditions and the rationale behind them is clear. However, a brief mention of why the first 3000ms of each simulation were discarded for analysis would be helpful.

Reviewer 2 Report

Comments and Suggestions for Authors

This paper uses a pre-defined computational model (developed by a separate group of researchers in a different context) to examine the mechanism of sustained increases in gamma power following sub-anesthetic doses of ketamine. The contention advanced in this paper is that this effect could underlie (at least in part) the therapeutic effects of sub-anesthetic ketamine in patients with depression.

While there are a number of interesting findings in this paper that are of interest to both researchers and clinicians, the work as presented has several significant limitations:

1. The computational model used in this study (Neymotin et al., 2011) is meant to simulate the functioning of the normal hippocampus. In fact, Neymotin et al.'s argument is that the disruption of theta modulation of gamma by ketamine in this model could explain the cognitive deficits seen in conditions such as schizophrenia, for which ketamine is often used as an experimental model (e.g., in animal research). Such a model cannot be used for the purpose of the current paper for two key reasons:

a. First of all, there is significant evidence of structural and functional alterations in the hippocampi of patients with depression. This includes reductions in hippocampal volume (Videbech & Ravnkilde, 2004; Amone et al., 2012; Santos et al., 2018), reduced volume of the CA3 network of the left hippocampus (Sun et al., 2023), and increased activity of the left hippocampus and parahippocampal gyrus (Ma et al., 2019). In order to effectively model the effects of ketamine in depression, one would need to incorporate these changes into the Neymotin et al. model "at baseline". Otherwise, what is being modelled is the effect of ketamine in "normal" hippocampus, not in the hippocampus of patients suffering from depression.

b. As mentioned above, Neymotin et al. were attempting to provide an explanation for the undesirable effects of ketamine (e.g., cognitive dysfunction). It is not clear how this can be equated to a desirable or therapeutic effect in depression.

To summarize the above two points briefly: the current paper, though providing interesting insights into the possible neural network effects of ketamine, cannot be considered relevant to depression or to a specific therapeutic effect of ketamine. It would be difficult to address this without extensively reworking the computational model (e.g., by reducing numbers and connections of CA3 cells, or by providing an abnormal increase in baseline activity) prior to testing the effects of ketamine.

2. The evidence that ketamine's antidepressant effects are associated with increased gamma power have been somewhat misstated in this paper. As per the most recent meta-analysis (Medeiros et al., 2023), ketamine's effect in depression appears to be associated with increased gamma power in the frontoparietal cortices. It is not clear how one gets from increased gamma power in the hippocampus to increased gamma power in these regions, as they are not related in a simple linear manner.

3. The relationship between changes in gamma power and depression - or possible antidepressant mechanisms - is a complex one. While some studies do find evidence of reduced gamma power in this condition (e.g., Jiang et al., 2020), others have found links between higher gamma power and greater severity of specific depressive symptoms, such as ruminations or even suicidal ideation (e.g., Arikan et al., 2019). The latter finding is hard to square with the authors' hypotheses regarding ketamine, as ketamine has well-documented anti-suicidal effects in MDD.

4. The authors have included AMPA synapses in their computational model, introducing an additional (but unavoidable) layer of complexity. The effect of AMPA receptor activation on gamma oscillations in CA3 is complex (see Li et al., 2023) and this becomes even more complex when trying to account for relative activities of at various subtypes of glutamate receptors with or without extraneous drug administration (see Klemz et al., 2022). This should be taken into account when discussing the current study's findings.

5. In the introduction, the link between synaptic plasticity and depression is highlighted. It would be valuable to explain how this relates to alterations in gamma power (e.g., is increased hippocampal synaptic plasticity associated with significant increases in gamma power)?

6. Various mechanisms of action for ketamine in depression have been proposed, involving effects not just at glutamate receptors but in interaction with other neurotransmitters, hormones and inflammatory mediators. While I agree that it would be nearly impossible to model all these without an exponential increase in computing power, it would be worth including these in the Discussion and explaining how the current study's findings fit alongside them.

In brief, I did not find any significant errors in the study methodology and results, which are the strongest part of the paper. However, given the conceptual concerns mentioned above, I think a certain degree of agnosticism about the usefulness of this particular mechanism in depression per se is in order, and this should be addressed through a more extensive literature review, a careful rewording of the title, introduction and conclusions, and an acknowledgement of the complexities involved in even a "simple" model of this sort.

Reviewer 3 Report

Comments and Suggestions for Authors

This work addresses an important topic regarding the mechanisms behind ketamine Mechanisms of Sustained Increases in γ Power following Subanaesthetic Ketamine in Depression.

The manuscript is highly informative and well-written

Some concerns about the clinical implication of the results, for example Ketamine is a highly addictive medication, does the study support its use in the future? Ketamine could posses a faster onset of action compared to the classical antidepressants used

Also, regarding the disinhibition hypothesis and the direct inhibition hypothesis discussed, the study did not investigate important proteins or genes accompanied by electrophysiological signatures, is this because of the computational model employed.

 The methods are nicely detailed

The results are well- presented

Round 2

Reviewer 2 Report

Comments and Suggestions for Authors

The revised manuscript has addressed the concerns raised in my earlier review. I have no further changes or corrections to suggest.